# ImageNet-trained CNNs are biased towards texture; increasing shape bias improves accuracy and robustness

**Robert Geirhos**
University of Tübingen & IMPRS-IS
robert.geirhos@bethgelab.org

**Patricia Rubisch**
University of Tübingen & U. of Edinburgh
p.rubisch@sms.ed.ac.uk

**Claudio Michaelis**
University of Tübingen & IMPRS-IS
claudio.michaelis@bethgelab.org

**Matthias Bethge**[*]
University of Tübingen
matthias.bethge@bethgelab.org

**Felix A. Wichmann**[*]
University of Tübingen
felix.wichmann@uni-tuebingen.de

**Wieland Brendel**[*]
University of Tübingen
wieland.brendel@bethgelab.org

## ABSTRACT

Convolutional Neural Networks (CNNs) are commonly thought to recognise objects by learning increasingly complex representations of object shapes. Some recent studies suggest a more important role of image textures. We here put these conflicting hypotheses to a quantitative test by evaluating CNNs and human observers on images with a texture-shape cue conflict. We show that ImageNet-trained CNNs are strongly biased towards recognising textures rather than shapes, which is in stark contrast to human behavioural evidence and reveals fundamentally different classification strategies. We then demonstrate that the same standard architecture (ResNet-50) that learns a texture-based representation on ImageNet is able to learn a shape-based representation instead when trained on 'Stylized-ImageNet', a stylized version of ImageNet. This provides a much better fit for human behavioural performance in our well-controlled psychophysical lab setting (nine experiments totalling 48,560 psychophysical trials across 97 observers) and comes with a number of unexpected emergent benefits such as improved object detection performance and previously unseen robustness towards a wide range of image distortions, highlighting advantages of a shape-based representation.

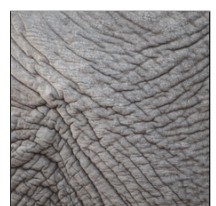 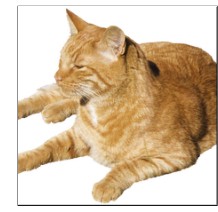 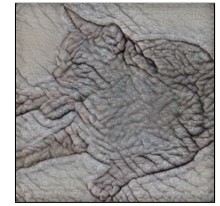

| | | |
|---|---|---|
| (a) Texture image | (b) Content image | (c) Texture-shape cue conflict |
| 81.4%  **Indian elephant** | 71.1%  **tabby cat** | 63.9%  **Indian elephant** |
| 10.3%  indri | 17.3%  grey fox | 26.4%  indri |
| 8.2%  black swan | 3.3%  Siamese cat | 9.6%  black swan |

Figure 1: Classification of a standard ResNet-50 of **(a)** a texture image (elephant skin: only texture cues); **(b)** a normal image of a cat (with both shape and texture cues), and **(c)** an image with a texture-shape cue conflict, generated by style transfer between the first two images.

---

[*]Joint senior authors

# 1 INTRODUCTION

How are Convolutional Neural Networks (CNNs) able to reach impressive performance on complex perceptual tasks such as object recognition (Krizhevsky et al., 2012) and semantic segmentation (Long et al., 2015)? One widely accepted intuition is that CNNs combine low-level features (e.g. edges) to increasingly complex shapes (such as wheels, car windows) until the object (e.g. car) can be readily classified. As Kriegeskorte (2015) puts it, "the network acquires complex knowledge about the kinds of shapes associated with each category. [...] High-level units appear to learn representations of shapes occurring in natural images" (p. 429). This notion also appears in other explanations, such as in LeCun et al. (2015): Intermediate CNN layers recognise "parts of familiar objects, and subsequent layers [...] detect objects as combinations of these parts" (p. 436). We term this explanation the *shape hypothesis*.

This hypothesis is supported by a number of empirical findings. Visualisation techniques like Deconvolutional Networks (Zeiler & Fergus, 2014) often highlight object parts in high-level CNN features.[1] Moreover, CNNs have been proposed as computational models of human shape perception by Kubilius et al. (2016), who conducted an impressive number of experiments comparing human and CNN shape representations and concluded that CNNs "implicitly learn representations of shape that reflect human shape perception" (p. 15). Ritter et al. (2017) discovered that CNNs develop a so-called "shape bias" just like children, i.e. that object shape is more important than colour for object classification (although see Hosseini et al. (2018) for contrary evidence). Furthermore, CNNs are currently the most predictive models for human ventral stream object recognition (e.g. Cadieu et al., 2014; Yamins et al., 2014); and it is well-known that object shape is the single most important cue for human object recognition (Landau et al., 1988), much more than other cues like size or texture (which may explain the ease at which humans recognise line drawings or millennia-old cave paintings).

On the other hand, some rather disconnected findings point to an important role of object textures for CNN object recognition. CNNs can still classify texturised images perfectly well, even if the global shape structure is completely destroyed (Gatys et al., 2017; Brendel & Bethge, 2019). Conversely, standard CNNs are bad at recognising object sketches where object shapes are preserved yet all texture cues are missing (Ballester & de Araújo, 2016). Additionally, two studies suggest that local information such as textures may actually be sufficient to "solve" ImageNet object recognition: Gatys et al. (2015) discovered that a linear classifier on top of a CNN's texture representation (Gram matrix) achieves hardly any classification performance loss compared to original network performance. More recently, Brendel & Bethge (2019) demonstrated that CNNs with explicitly constrained receptive field sizes throughout all layers are able to reach surprisingly high accuracies on ImageNet, even though this effectively limits a model to recognising small local patches rather than integrating object parts for shape recognition. Taken together, it seems that local textures indeed provide sufficient information about object classes—ImageNet object recognition *could*, in principle, be achieved through texture recognition alone. In the light of these findings, we believe that it is time to consider a second explanation, which we term the *texture hypothesis*: in contrast to the common assumption, object textures are more important than global object shapes for CNN object recognition.

Resolving these two contradictory hypotheses is important both for the deep learning community (to increase our understanding of neural network decisions) as well as for the human vision and neuroscience communities (where CNNs are being used as computational models of human object recognition and shape perception). In this work we aim to shed light on this debate with a number of carefully designed yet relatively straightforward experiments. Utilising style transfer (Gatys et al., 2016), we created images with a texture-shape cue conflict such as the cat shape with elephant texture depicted in Figure 1c. This enables us to quantify texture and shape biases in both humans and CNNs. To this end, we perform nine comprehensive and careful psychophysical experiments comparing humans against CNNs on exactly the same images, totalling 48,560 psychophysical trials across 97 observers. These experiments provide behavioural evidence in favour of the texture hypothesis: A cat with an elephant texture is an elephant to CNNs, and still a cat to humans. Beyond quantifying existing biases, we subsequently present results for our two other main contributions:

---

[1]To avoid any confusion caused by different meanings of the term 'feature', we consistently use it to refer to properties of CNNs (learned features) rather than to object properties (such as colour). When referring to physical objects, we use the term 'cue' instead.

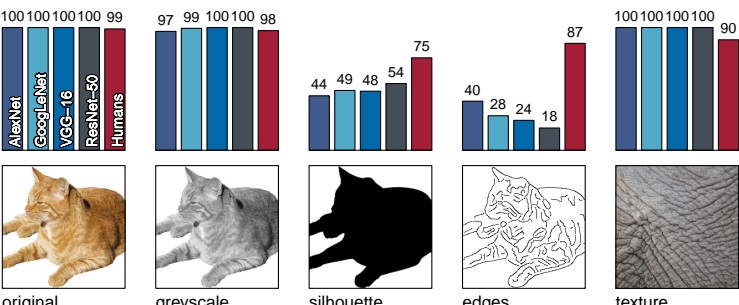

Figure 2: Accuracies and example stimuli for five different experiments without cue conflict.

changing biases, and discovering emergent benefits of changed biases. We show that the texture bias in standard CNNs can be overcome and changed towards a shape bias if trained on a suitable data set. Remarkably, networks with a higher shape bias are inherently more robust to many different image distortions (for some even reaching or surpassing human performance, *despite never being trained on any of them*) and reach higher performance on classification and object recognition tasks.

## 2 METHODS

In this section we outline the core elements of paradigm and procedure. Extensive details to facilitate replication are provided in the Appendix. Data, code and materials are available from this repository: `https://github.com/rgeirhos/texture-vs-shape`

### 2.1 PSYCHOPHYSICAL EXPERIMENTS

All psychophysical experiments were conducted in a well-controlled psychophysical lab setting and follow the paradigm of Geirhos et al. (2018), which allows for direct comparisons between human and CNN classification performance on exactly the same images. Briefly, in each trial participants were presented a fixation square for 300 ms, followed by a 300 ms presentation of the stimulus image. After the stimulus image we presented a full-contrast pink noise mask ($1/f$ spectral shape) for 200 ms to minimise feedback processing in the human visual system and to thereby make the comparison to feedforward CNNs as fair as possible. Subsequently, participants had to choose one of 16 entry-level categories by clicking on a response screen shown for 1500 ms. On this screen, icons of all 16 categories were arranged in a $4 \times 4$ grid. Those categories were `airplane`, `bear`, `bicycle`, `bird`, `boat`, `bottle`, `car`, `cat`, `chair`, `clock`, `dog`, `elephant`, `keyboard`, `knife`, `oven` and `truck`. Those are the so-called "16-class-ImageNet" categories introduced in Geirhos et al. (2018).

The same images were fed to four CNNs pre-trained on standard ImageNet, namely AlexNet (Krizhevsky et al., 2012), GoogLeNet (Szegedy et al., 2015), VGG-16 (Simonyan & Zisserman, 2015) and ResNet-50 (He et al., 2015). The 1,000 ImageNet class predictions were mapped to the 16 categories using the WordNet hierarchy (Miller, 1995)—e.g. ImageNet category `tabby cat` would be mapped to `cat`. In total, the results presented in this study are based on 48,560 psychophysical trials and 97 participants.

### 2.2 DATA SETS (PSYCHOPHYSICS)

In order to assess texture and shape biases, we conducted six major experiments along with three control experiments, which are described in the Appendix. The first five experiments (samples visualised in Figure 2) are simple object recognition tasks with the only difference being the image features available to the participant:

**Original** 160 natural colour images of objects (10 per category) with white background.

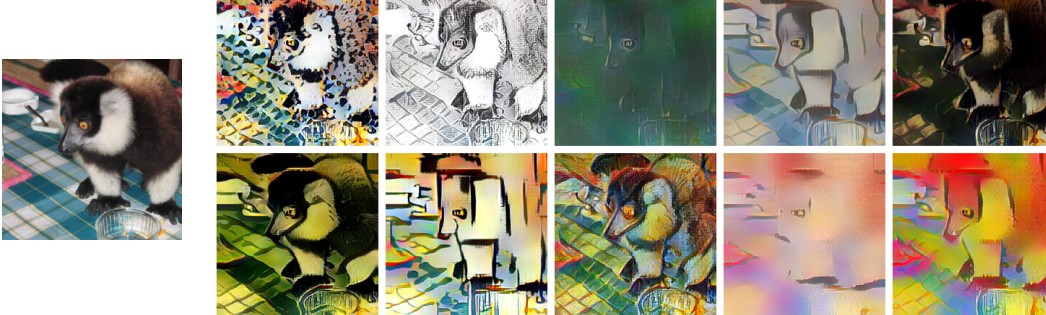

Figure 3: Visualisation of Stylized-ImageNet (SIN), created by applying AdaIN style transfer to ImageNet images. Left: randomly selected ImageNet image of class `ring-tailed lemur`. Right: ten examples of images with content/shape of left image and style/texture from different paintings. After applying AdaIN style transfer, local texture cues are no longer highly predictive of the target class, while the global shape tends to be retained. Note that within SIN, every source image is stylized only once.

**Greyscale**    Images from *Original* data set converted to greyscale using `skimage.color.rgb2gray`. For CNNs, greyscale images were stacked along the colour channel.

**Silhouette**    Images from *Original* data set converted to silhouette images showing an entirely black object on a white background (see Appendix A.6 for procedure).

**Edges**    Images from *Original* data set converted to an edge-based representation using `Canny edge extractor` implemented in MATLAB.

**Texture**    48 natural colour images of textures (3 per category). Typically the textures consist of full-width patches of an animal (e.g. skin or fur) or, in particular for man-made objects, of images with many repetitions of the same objects (e.g. many bottles next to each other, see Figure 7 in the Appendix).

It is important to note that we only selected object and texture images that were correctly classified by all four networks. This was made to ensure that our results in the sixth experiment on cue conflicts, which is most decisive in terms of the shape vs texture hypothesis, are fully interpretable. In the cue conflict experiment we present images with contradictory features (see Figure 1) but still ask the participant to assign a single class. Note that the instructions to human observers were entirely neutral w.r.t. shape or texture ("click on the object category that you see in the presented image; guess if unsure. There is no right or wrong answer, we are interested in your subjective impression").

**Cue conflict**    Images generated using iterative style transfer (Gatys et al., 2016) between an image of the *Texture* data set (as style) and an image from the *Original* data set (as content). We generated a total of 1280 cue conflict images (80 per category), which allows for presentation to human observers within a single experimental session.

We define "silhouette" as the bounding contour of an object in 2D (i.e., the outline of object segmentation). When mentioning "object shape", we use a definition that is broader than just the silhouette of an object: we refer to the set of contours that describe the 3D form of an object, i.e. including those contours that are not part of the silhouette. Following Gatys et al. (2017), we define "texture" as an image (region) with spatially stationary statistics. Note that on a very local level, textures (according to this definition) can have non-stationary elements (such as a local shape): e.g. a single bottle clearly has non-stationary statistics, but many bottles next to each other are perceived as a texture: "things" become "stuff" (Gatys et al., 2017, p. 178). For an example of a "bottle texture" see Figure 7.

## 2.3 STYLIZED-IMAGENET

Starting from ImageNet we constructed a new data set (termed Stylized-ImageNet or SIN) by stripping every single image of its original texture and replacing it with the style of a randomly selected painting through AdaIN style transfer (Huang & Belongie, 2017) (see examples in Figure 3) with a stylization coefficient of $\alpha = 1.0$. We used Kaggle's `Painter by Numbers` data set[2] as a style source due to its large style variety and size (79,434 paintings). We used AdaIN fast style transfer rather than iterative stylization (e.g. Gatys et al., 2016) for two reasons: Firstly, to ensure that training on SIN and testing on cue conflict stimuli is done using different stylization techniques, such that the results do not rely on a single stylization method. Secondly, to enable stylizing entire ImageNet, which would take prohibitively long with an iterative approach. We provide code to create Stylized-ImageNet here:
`https://github.com/rgeirhos/Stylized-ImageNet`

## 3 RESULTS

### 3.1 TEXTURE VS SHAPE BIAS IN HUMANS AND IMAGENET-TRAINED CNNS

Almost all object and texture images (*Original* and *Texture* data set) were recognised correctly by both CNNs and humans (Figure 2). Greyscale versions of the objects, which still contain both shape and texture, were recognised equally well. When object outlines were filled in with black colour to generate a silhouette, CNN recognition accuracies were much lower than human accuracies. This was even more pronounced for edge stimuli, indicating that human observers cope much better with images that have little to no texture information. One confound in these experiments is that CNNs tend not to cope well with domain shifts, i.e. the large change in image statistics from natural images (on which the networks have been trained) to sketches (which the networks have never seen before).

We thus devised a cue conflict experiment that is based on images with a natural statistic but contradicting texture and shape evidence (see Methods). Participants and CNNs have to classify the images based on the features (shape or texture) that they most rely on. The results of this experiment are visualised in Figure 4. Human observers show a striking bias towards responding with the shape category (95.9% of correct decisions).[3] This pattern is reversed for CNNs, which show a clear bias towards responding with the texture category (VGG-16: 17.2% shape vs. 82.8% texture; GoogLeNet: 31.2% vs. 68.8%; AlexNet: 42.9% vs. 57.1%; ResNet-50: 22.1% vs. 77.9%).

### 3.2 OVERCOMING THE TEXTURE BIAS OF CNNS

The psychophysical experiments suggest that ImageNet-trained CNNs, but not humans, exhibit a strong texture bias. One reason might be the training task itself: from Brendel & Bethge (2019) we know that ImageNet can be solved to high accuracy using only local information. In other words, it might simply suffice to integrate evidence from many local texture features rather than going through the process of integrating and classifying global shapes. In order to test this hypothesis we train a ResNet-50 on our Stylized-ImageNet (SIN) data set in which we replaced the object-related local texture information with the uninformative style of randomly selected artistic paintings.

A standard ResNet-50 trained and evaluated on Stylized-ImageNet (SIN) achieves 79.0% top-5 accuracy (see Table 1). In comparison, the same architecture trained and evaluated on ImageNet (IN) achieves 92.9% top-5 accuracy. This performance difference indicates that SIN is a much harder task than IN since textures are no longer predictive, but instead a nuisance factor (as desired). Intriguingly, ImageNet features generalise poorly to SIN (only 16.4% top-5 accuracy); yet features learned on SIN generalise very well to ImageNet (82.6% top-5 accuracy without any fine-tuning).

In order to test wheter local texture features are still sufficient to "solve" SIN we evaluate the performance of so-called *BagNets*. Introduced recently by Brendel & Bethge (2019), BagNets have a ResNet-50 architecture but their maximum receptive field size is limited to $9 \times 9$, $17 \times 17$ or $33 \times 33$

---

[2]`https://www.kaggle.com/c/painter-by-numbers/` (accessed on March 1, 2018).

[3]It is important to note that a substantial fraction of the images (automatically generated with style transfer between randomly selected object image and texture image) seemed hard to recognise for both humans and CNNs, as depicted by the fraction of incorrect classification choices in Figure 4.

Figure 4: Classification results for human observers (red circles) and ImageNet-trained networks AlexNet (purple diamonds), VGG-16 (blue triangles), GoogLeNet (turquoise circles) and ResNet-50 (grey squares). Shape vs. texture biases for stimuli with cue conflict (sorted by human shape bias). Within the responses that corresponded to either the correct texture or correct shape category, the fractions of texture and shape decisions are depicted in the main plot (averages visualised by vertical lines). On the right side, small barplots display the proportion of correct decisions (either texture or shape correctly recognised) as a fraction of all trials. Similar results for ResNet-152, DenseNet-121 and Squeezenet1_1 are reported in the Appendix, Figure 13.

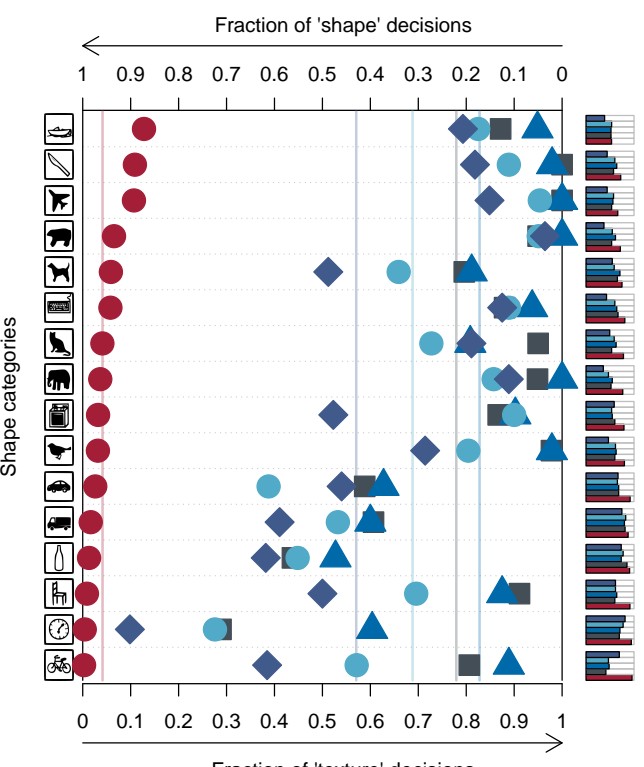

pixels. This precludes BagNets from learning or using any long-range spatial relationships for classification. While these restricted networks can reach high accuracies on ImageNet, they are unable to achieve the same on SIN, showing dramatically reduced performance with smaller receptive field sizes (such as 10.0% top-5 accuracy on SIN compared to 70.0% on ImageNet for a BagNet with receptive field size of $9 \times 9$ pixels). This is a clear indication that the SIN data set we propose does actually remove local texture cues, forcing a network to integrate long-range spatial information.

Most importantly, the SIN-trained ResNet-50 shows a much stronger shape bias in our cue conflict experiment (Figure 5), which increases from 22% for a IN-trained model to 81%. In many categories the shape bias is almost as strong as for humans.

## 3.3 ROBUSTNESS AND ACCURACY OF SHAPE-BASED REPRESENTATIONS

Does the increased shape bias, and thus the shifted representations, also affect the performance or robustness of CNNs? In addition to the IN- and SIN-trained ResNet-50 architecture we here additionally analyse two joint training schemes:

- Training jointly on SIN and IN.
- Training jointly on SIN and IN with fine-tuning on IN. We refer to this model as *Shape-ResNet*.

| architecture | IN→IN | IN→SIN | SIN→SIN | SIN→IN |
|---|---|---|---|---|
| ResNet-50 | 92.9 | 16.4 | 79.0 | 82.6 |
| BagNet-33 (mod. ResNet-50) | 86.4 | 4.2 | 48.9 | 53.0 |
| BagNet-17 (mod. ResNet-50) | 80.3 | 2.5 | 29.3 | 32.6 |
| BagNet-9 (mod. ResNet-50) | 70.0 | 1.4 | 10.0 | 10.9 |

Table 1: Stylized-ImageNet cannot be solved with texture features alone. Accuracy comparison (in percent; top-5 on validation data set) of a standard ResNet-50 with Bag of Feature networks (BagNets) with restricted receptive field sizes of 33×33, 17×17 and 9×9 pixels. Arrows indicate: train data→test data, e.g. IN→SIN means training on ImageNet and testing on Stylized-ImageNet.

Figure 5: Shape vs. texture biases for stimuli with a texture-shape cue conflict after training ResNet-50 on Stylized-ImageNet (orange squares) and on ImageNet (grey squares). Plotting conventions and human data (red circles) for comparison are identical to Figure 4. Similar results for other networks are reported in the Appendix, Figure 11.

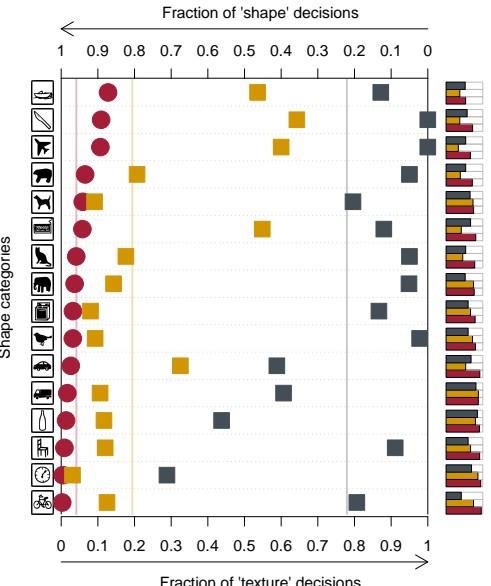

| name | training | fine-tuning | top-1 IN accuracy (%) | top-5 IN accuracy (%) | Pascal VOC mAP50 (%) | MS COCO mAP50 (%) |
|------|----------|-------------|----------------------|----------------------|----------------------|-------------------|
| vanilla ResNet | IN | - | 76.13 | 92.86 | 70.7 | 52.3 |
| | SIN | - | 60.18 | 82.62 | 70.6 | 51.9 |
| | SIN+IN | - | 74.59 | 92.14 | 74.0 | 53.8 |
| Shape-ResNet | SIN+IN | IN | **76.72** | **93.28** | **75.1** | **55.2** |

Table 2: Accuracy comparison on the ImageNet (IN) validation data set as well as object detection performance (mAP50) on PASCAL VOC 2007 and MS COCO. All models have an identical ResNet-50 architecture. Method details reported in the Appendix, where we also report similar results for ResNet-152 (Table 4).

We then compared these models with a vanilla ResNet-50 on three experiments: (1) classification performance on IN, (2) transfer to Pascal VOC 2007 and (3) robustness against image perturbations.

**Classification performance**  Shape-ResNet surpasses the vanilla ResNet in terms of top-1 and top-5 ImageNet validation accuracy as reported in Table 2. This indicates that SIN may be a useful data augmentation on ImageNet that can improve model performance without any architectural changes.

**Transfer learning**  We tested the representations of each model as backbone features for Faster R-CNN (Ren et al., 2017) on Pascal VOC 2007 and MS COCO. Incorporating SIN in the training data substantially improves object detection performance from 70.7 to 75.1 mAP50 (52.3 to 55.2 mAP50 on MS COCO) as shown in Table 2. This is in line with the intuition that for object detection, a shape-based representation is more beneficial than a texture-based representation, since the ground truth rectangles encompassing an object are by design aligned with global object shape.

**Robustness against distortions**  We systematically tested how model accuracies degrade if images are distorted by uniform or phase noise, contrast changes, high- and low-pass filtering or eidolon perturbations.[4] The results of this comparison, including human data for reference, are visualised in Figure 6. While lacking a few percent accuracy on undistorted images, the SIN-trained network outperforms the IN-trained CNN on almost all image manipulations. (Low-pass filtering / blurring is the only distortion type on which SIN-trained networks are more susceptible, which might be due to the over-representation of high frequency signals in SIN through paintings and the reliance on

---

[4]Our comparison encompasses all distortions reported by Geirhos et al. (2018) with more than five different levels of signal strength. Data from human observers included with permission from the authors (see appendix).

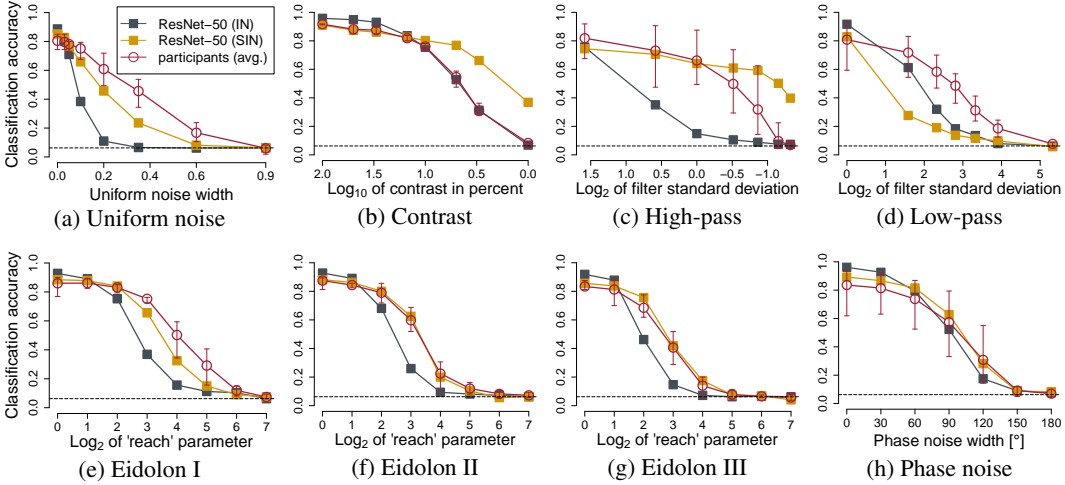

Figure 6: Classification accuracy on parametrically distorted images. ResNet-50 trained on Stylized-ImageNet (SIN) is more robust towards distortions than the same network trained on ImageNet (IN).

sharp edges.) The SIN-trained ResNet-50 approaches human-level distortion robustness—*despite never seeing any of the distortions during training*.

Furthermore, we provide robustness results for our models tested on ImageNet-C, a comprehensive benchmark of 15 different image corruptions (Hendrycks & Dietterich, 2019), in Table 5 of the Appendix. Training jointly on SIN and IN leads to strong improvements for 13 corruption types (Gaussian, Shot and Impulse noise; Defocus, Glas and Motion blur; Snow, Frost and Fog weather types; Contrast, Elastic, Pixelate and JPEG digital corruptions). This substantially reduces overall corruption error from 76.7 for a vanilla ResNet-50 to 69.3. Again, none of these corruption types were explicitly part of the training data, reinforcing that incorporating SIN in the training regime improves model robustness in a very general way.

## 4 DISCUSSION

As noted in the Introduction, there seems to be a large discrepancy between the common assumption that CNNs use increasingly complex shape features to recognise objects and recent empirical findings which suggest a crucial role of object textures instead. In order to explicitly probe this question, we utilised style transfer (Gatys et al., 2016) to generate images with conflicting shape and texture information. On the basis of extensive experiments on both CNNs and human observers in a controlled psychophysical lab setting, we provide evidence that unlike humans, ImageNet-trained CNNs tend to classify objects according to local textures instead of global object shapes. In combination with previous work which showed that changing other major object dimensions such as colour (Geirhos et al., 2018) and object size relative to the context (Eckstein et al., 2017) do not have a strong detrimental impact on CNN recognition performance, this highlights the special role that local cues such as textures seem to play in CNN object recognition.

Intriguingly, this offers an explanation for a number of rather disconnected findings: CNNs match texture appearance for humans (Wallis et al., 2017), and their predictive power for neural responses along the human ventral stream appears to be largely due to human-like texture representations, but not human-like contour representations (Laskar et al., 2018; Long & Konkle, 2018). Furthermore, texture-based generative modelling approaches such as style transfer (Gatys et al., 2016), single image super-resolution (Gondal et al., 2018) as well as static and dynamic texture synthesis (Gatys et al., 2015; Funke et al., 2017) all produce excellent results using standard CNNs, while CNN-based shape transfer seems to be very difficult (Gokaslan et al., 2018). CNNs can still recognise images with scrambled shapes (Gatys et al., 2017; Brendel & Bethge, 2019), but they have much more difficulties recognising objects with missing texture information (Ballester & de Araújo, 2016; Yu et al., 2017). Our hypothesis might also explain why an image segmentation model trained on a database of synthetic texture images transfers to natural images and videos (Ustyuzhaninov et al.,

2018). Beyond that, our results show marked behavioural differences between ImageNet-trained CNNs and human observers. While both human and machine vision systems achieve similarly high accuracies on standard images (Geirhos et al., 2018), our findings suggest that the underlying classification strategies might actually be very different. This is problematic, since CNNs are being used as computational models for human object recognition (e.g. Cadieu et al., 2014; Yamins et al., 2014).

In order to reduce the texture bias of CNNs we introduced Stylized-ImageNet (SIN), a data set that removes local cues through style transfer and thereby forces networks to go beyond texture recognition. Using this data set, we demonstrated that a ResNet-50 architecture can indeed learn to recognise objects based on object shape, revealing that the texture bias in current CNNs is not by design but induced by ImageNet training data. This indicates that standard ImageNet-trained models may be taking a "shortcut" by focusing on local textures, which could be seen as a version of Occam's razor: If textures are sufficient, why should a CNN learn much else? While texture classification may be easier than shape recognition, we found that shape-based features trained on SIN generalise well to natural images.

Our results indicate that a more shape-based representation can be beneficial for recognition tasks that rely on pre-trained ImageNet CNNs. Furthermore, while ImageNet-trained CNNs generalise poorly towards a wide range of image distortions (e.g. Dodge & Karam, 2017; Geirhos et al., 2017; 2018), our ResNet-50 trained on Stylized-ImageNet often reaches or even surpasses human-level robustness (without ever being trained on the specific image degradations). This is exciting because Geirhos et al. (2018) showed that networks trained on specific distortions in general do not acquire robustness against other unseen image manipulations. This emergent behaviour highlights the usefulness of a shape-based representation: While local textures are easily distorted by all sorts of noise (including those in the real world, such as rain and snow), the object shape remains relatively stable. Furthermore, this finding offers a compellingly simple explanation for the incredible robustness of humans when coping with distortions: a shape-based representation.

## 5 CONCLUSION

In summary, we provided evidence that machine recognition today overly relies on object textures rather than global object shapes as commonly assumed. We demonstrated the advantages of a shape-based representation for robust inference (using our Stylized-ImageNet data set[5] to induce such a representation in neural networks). We envision our findings as well as our openly available model weights, code and behavioural data set (49K trials across 97 observers)[6] to achieve three goals: Firstly, an improved understanding of CNN representations and biases. Secondly, a step towards more plausible models of human visual object recognition. Thirdly, a useful starting point for future undertakings where domain knowledge suggests that a shape-based representation may be more beneficial than a texture-based one.

ACKNOWLEDGMENTS

This work has been funded, in part, by the German Research Foundation (DFG; Sachbeihilfe Wi 2103/4-1 and SFB 1233 on "Robust Vision"). The authors thank the International Max Planck Research School for Intelligent Systems (IMPRS-IS) for supporting R.G. and C.M.; M.B. acknowledges support by the Centre for Integrative Neuroscience Tübingen (EXC 307) and by the Intelligence Advanced Research Projects Activity (IARPA) via Department of Interior/Interior Business Center (DoI/IBC) contract number D16PC00003.

We would like to thank Dan Hendrycks for providing the results of Table 5 (corruption robustness of our models on ImageNet-C). Furthermore, we would like to express our gratitude towards Alexander Ecker, Leon Gatys, Tina Gauger, Silke Gramer, Heike König, Jonas Rauber, Steffen Schneider, Heiko Schütt, Tom Wallis and Uli Wannek for support and/or useful discussions.

---

[5]Available from `https://github.com/rgeirhos/Stylized-ImageNet`
[6]Available from `https://github.com/rgeirhos/texture-vs-shape`

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

## A  APPENDIX

### A.1  REPRODUCIBILITY & ACCESS TO CODE / MODELS / DATA

In this Appendix, we report experimental details for human and CNN experiments. All trained model weights reported in this paper as well as our human behavioural data set (48,560 psychophysical trials across 97 observers) are openly available from this repository:
https://github.com/rgeirhos/texture-vs-shape

### A.2  PROCEDURE

We followed the paradigm of Geirhos et al. (2018) for maximal comparability. A trial consisted of 300 ms presentation of a fixation square and a 200 ms presentation of the stimulus image, which was followed by a full-contrast pink noise mask (1/$f$ spectral shape) of the same size lasting for 200 ms. Participants had to choose one of 16 entry-level categories by clicking on a response screen shown for 1500 ms. On this screen, icons of all 16 categories were arranged in a $4 \times 4$ grid. The experiments were not self-paced and therefore one trial always lasted 2200 ms (300 ms + 200 ms + 200 ms + 1500 ms = 2200 ms). The necessary time to complete an experiment with 1280 stimuli was 47 minutes, for 160 stimuli six minutes, and for 48 stimuli two minutes. In the experiments with 1280 trials, observers were given the possibility of taking a brief break after every block of 256 trials (five blocks in total).

As preparation, participants were shown the response screen prior to an experiment and were asked to name all 16 categories in order to get an overview over the possible stimuli categories and to make sure that all categories were clear from the beginning. They were instructed to click on the category they believed was presented. Responses through clicking on a response screen could be changed within the 1500 ms response interval, only the last entered response was counted as the answer. Prior to the real experiment a practice session was performed for the participants to get used to the time course of the experiment and the position of category items on the response screen. This screen was shown for an additional 300 ms in order to provide feedback and indicate whether the entered answer was incorrect. In that case, a short low beep sound occurred and the correct category was highlighted by setting its background to white. The practice session consisted of 320 trials. After 160 trials the participants had the chance to take a short break. In the break, their performance of the first block was shown on the screen along the percentage of trials where no answer was entered. After the practice blocks, observers were shown an example image of the manipulation (not used in the experiment) to minimise surprise. Images used in the practice session were natural images from 16-class-ImageNet (Geirhos et al., 2018), hence there was no overlap with images or manipulations used in the experiments.

### A.3  APPARATUS

Observers were shown the $224 \times 224$ pixels stimuli in a dark cabin on a 22", 120 Hz VIEWPixx LCD monitor (VPixx Technologies, Saint-Bruno, Canada). The screen of size $484 \times 302$ mm corresponds to $1920 \times 1200$ pixels, although stimuli were only presented foveally at the center of the screen ($3 \times 3$ degrees of visual angle at a viewing distance of 107 cm) while the background was set to a grey value of 0.7614 in the $[0, 1]$ range, the average greyscale value of all stimuli used in the original experiment. Participants used a chin rest to keep their head position static during an experiment. Stimulus presentation was conducted with the Psychophysics Toolbox (version 3.0.12) in MATLAB (Release 2016a, The MathWorks, Inc., Natick, Massachusetts, United States) using a 12-core desktop computer (AMD HD7970 graphics card "Tahiti" by AMD, Sunnyvale, California, United States) running Kubuntu 14.04 LTS. Participants clicked on a response screen, showing an

| experiment | instruction | # p. | #♀ | #♂ | age range | mean age | # stimuli | rt |
|---|---|---|---|---|---|---|---|---|
| original | neutral | 5 | 5 | 0 | 21–27 | 24.2 | 160 | 772 |
| greyscale | neutral | 5 | 4 | 1 | 20–26 | 23.4 | 160 | 811 |
| texture | neutral | 5 | 2 | 3 | 23–36 | 29.0 | 48 | 769 |
| silhouette | neutral | 10 | 9 | 1 | 21–37 | 24.1 | 160 | 861 |
| edge | neutral | 10 | 6 | 4 | 18–30 | 23.0 | 160 | 791 |
| cue conflict | neutral | 10 | 7 | 3 | 20–29 | 23.0 | 1280 | 828 |
| cue conflict control | texture | 10 | 5 | 5 | 23–32 | 26.6 | 1280 | 942 |
| cue conflict control | shape | 10 | 9 | 1 | 18–25 | 21.8 | 1280 | 827 |
| filled silhouette | neutral | 32 | 22 | 10 | 18–30 | 22.3 | 160 | 881 |
| overall | | 97 | 69 | 28 | 18–37 | 23.5 | 48,560 trials | 857 |

Table 3: Characteristics of human participants (p.) across experiments. The symbol '#' refers to "number of"; 'rt' stands for "median reaction time (ms)" in an experiment.

iconic representation of all of the 16 object categories as reported in Geirhos et al. (2018), with a normal computer mouse.

## A.4 PARTICIPANTS

In total, 97 human observers participated in the study. For a detailed overview about how they were distributed across experiments see Table 3. No observer participated in more than one experiment, and all participants reported normal or corrected-to-normal vision. Observers participating in experiments with a cue conflict manipulation were paid € 10 per hour or gained course credit. Observers measured in all other experiments (with a clear ground truth category) were able to earn an additional bonus up to € 5 or equivalent further course credit based on their performance. This motivation scheme was applied to ensure reliable answer rates, and explained to observers in advance. Participant bonus, in these cases, was calculated as follows: The base level with a bonus of € 0 was set to 50% accuracy. For every additional 5% of accuracy, participants gained a € 0.50 bonus. This means that with a performance above 95%, an observer was able to gain the full bonus of € 5 or equivalent course credit. Overall, we took the following steps to prevent low quality human data: 1., using a controlled lab environment instead of an online crowdsourcing platform; 2. the payment motivation scheme as explained above; 3. displaying observer performance on the screen at regular intervals during the practice session; and 4. splitting longer experiments into five blocks, where participants could take a break in between blocks.

## A.5 CNN MODELS & TRAINING DETAILS

**ResNet-50** We used a standard ResNet-50 architecture from PyTorch (Paszke et al., 2017), the `torchvision.models.resnet50` implementation. For the comparison against BagNets reported in Table 1, results for IN training correspond to a ResNet-50 pre-trained on ImageNet without any modifications (model weights from `torchvision.models`). Reported results for SIN training correspond to the same architecture trained on SIN for 60 epochs with Stochastic Gradient Descent (`torch.optim.SGD`) using a momentum term of 0.9, weight decay (1e-4) and a learning rate of 0.1 which was multiplied by a factor of 0.1 after 20 and 40 epochs of training. We used a batch size of 256. This SIN-trained model is the same model that is reported in Figures 5 and 6 as well as in Table 2. In the latter, this corresponds to the second row (training performed on SIN, no fine-tuning on ImageNet). For the model reported in the third row, training was jointly performed on SIN and on IN. This means that both training data sets were treated as one big data set (exactly twice the size of the IN training data set), on which training was performed for 45 epochs with identical hyperparameters as described above, except that the initial learning rate of 0.1 was multiplied by 0.1 after 15 and 30 epochs. The weights of this model were then used to initialise the model reported in the fourth row of Table 2, which was fine-tuned for 60 epochs on ImageNet (identical hyperparameters except that the initial learning rate of 0.01 was multiplied by 0.1 after 30 epochs). We compared training models from scratch versus starting from an ImageNet-pretrained model. Empirically, using features pre-trained on ImageNet led to better results across experiments, which is

why we used ImageNet pre-training throughout experiments and models (for both ResNet-50 and restricted ResNet-50 models).

**BagNets**  Model weights (pre-trained on ImageNet) and architectures for BagNets (results reported in Table 1) were kindly provided by Brendel & Bethge (2019). For SIN training, identical settings as for the SIN-trained ResNet-50 were used to ensure comparability (training for 60 epochs with SGD and identical hyperparameters as reported above).

**Faster R-CNN**  We used the Faster R-CNN implementation from `https://github.com/jwyang/faster-rcnn.pytorch` (commit 21f28986) with all hyperparameters kept at default. The only changes we made to the model is replacing the encoder with ResNet-50 (respectively ResNet-152 for the results in Table 4) and applying custom input whitening. For Pascal VOC 2007 we trained the model for 7 epochs with a batch size of 1, a learning rate of 0.001 and a learning rate decay step after epoch 5. Images were resized to have a short edge of 600 pixels. For MS COCO we trained the same model on the 2017 train/val split for training and testing respectively. We trained for 6 epochs with a batch size of 16 on 8 GPUs employing a learning rate of 0.02 and a decay step after 4 epochs. Images were resized to have a short edge of 800 pixels.

**Pre-trained AlexNet, GoogLeNet, VGG-16**  We used AlexNet (Krizhevsky et al., 2012), GoogLeNet (Szegedy et al., 2015) and VGG-16 (Simonyan & Zisserman, 2015) for the evaluation reported in Figure 4. Evaluation was performed using Caffe (Jia et al., 2014). Network weights (training on ImageNet) were obtained from `https://github.com/BVLC/caffe/wiki/Model-Zoo` (AlexNet & GoogLeNet) and `http://www.robots.ox.ac.uk/` (VGG-16).

**ResNet-101 pre-trained on Open Images V2**  For our comparison of biases in ImageNet vs. OpenImages (Figure 13 right) the ResNet-101 pretrained on Open Images V2 (Krasin et al., 2017) was used. It was obtained from `https://github.com/openimages/dataset/blob/master/READMEV2.md` along with the inference code provided by the authors. In order to map predictions to the 16 classes, we used the parameters $top\_k = 100000$ and $score\_threshold = 0.0$ to obtain as all predictions, and then mapped the responses to our 16 classes using the provided label map. 15 out of our 16 classes are classes in Open Images as well; the remaining class `keyboard` was mapped to Open Images class `computer keyboard` (in this case, Open Images makes a finer distinction to separate musical keyboards from computer keyboards).

**ResNet-101, ResNet-152, DenseNet-121, SqueezeNet1_1**  For the comparison to other models pre-trained on ImageNet (Figure 13 left), we evaluated the pre-trained networks provided by `torchvision.models`.

**Training AlexNet, VGG-16 on SIN**  For the evaluation of model biases after training on SIN (Figure 11), we obtained the model architectures from `torchvision.models` and trained the networks under identical circumstances as ResNet-50. This includes identical hyperparameter settings, except for the learning rate. The learning rate for AlexNet was set to 0.001 and for VGG-16 to 0.01 initially; both learning rates were multiplied by 0.1 after 20 and 40 epochs of training (60 epochs in total).

## A.6  Image manipulations and image database

In total, we conducted nine different experiments. Here is an overview of the images and / or image manipulations for all of them. All images were saved in the `png` format and had a size of $224 \times 224$ pixels. Original, texture and cue conflict images are visualised in Figure 7.

**Original experiment**  This experiment consisted of 160 coloured images, 10 per category. All of them had a single, unmanipulated object (belonging to one category) in front of a white background. This white background was especially important since these stimuli were being used as content images for style transfer, and we thus made sure that the background was neutral to produce better style transfer results. The images for this experiment as well as for the texture experiment

described below were carefully selected using Google advanced image search with the criteria "labelled for noncommercial reuse with modification (free to use, share and modify)" and the search term "<entity> white background" (original) or "<entity> texture" (texture). In some cases where this did not lead to sufficient results, we used images from the ImageNet validation data set which were manually modified to have a white background if necessary. We made sure that both the images from this experiment as well as the texture images were all correctly recognised by all four pre-trained CNNs (if an image was not correctly recognised, we replaced it by another one). This was used to ensure that our results for cue conflict experiments are fully interpretable: if, e.g., a texture image was not correctly recognised by CNNs, there would be no point in using it as a texture (style) source for style transfer.

**Greyscale experiment** This experiment used the same images as the original experiment with the difference that they were converted to greyscale using `skimage.color.rgb2gray`. For CNNs, greyscale images were stacked three times along the colour channel.

**Silhouette experiment** The images from the original experiment were transformed into silhouette images showing an entirely black object on a white background. We used the following transformation procedure: First, images were converted to `bmp` using command line utility (`convert`). They were then converted to `svg` using `potrace`, and then to `png` using `convert` again. Since an entirely automatic binarization pipeline is not feasible (it takes domain knowledge to understand that a car wheel should, but a doughnut should not be filled with black colour), we then manually checked every single image and adapted the silhouette using `GIMP` if necessary.

**Edge experiment** The stimuli shown in this condition were generated by applying the "Canny" edge extractor implemented in MATLAB (Release 2016a, The MathWorks, Inc., Natick, Massachusetts, United States) to the images used in the original experiment. No further manipulations were performed on this data set. This line of code was used to detect edges and generate the stimuli used in this experiment:

```
imwrite(1-edge(imgaussfilt(rgb2gray(imread(filename)), 2),
'Canny'), targetFilename);
```

**Texture experiment** Images were selected using the procedure outlined above for the original experiment. Some objects have a fairly stationary texture (e.g. animals), which makes it easy to find texture images for them. For the more difficult case (e.g. man-made objects), we made use of the fact that every object can become a texture if it is used not in isolation, but rather in a clutter of many objects of the same kind (e.g. Gatys et al., 2017). That is, for a `bottle` texture we used images with many bottles next to each other (as visualised in Figure 7).

**Cue conflict experiment** This experiment used images with a texture-shape cue conflict. They were generated using iterative style transfer (Gatys et al., 2016) between a texture image (from the texture experiment described above) and a content image (from the original experiment) each. While 48 texture images and 160 content images would allow for a total of $48 \times 160 = 7680$ cue conflict images (480 per category), we used a balanced subset of 1280 images instead (80 per category), which allows for presentation to human observers within a single experimental session. The procedure for selecting the style and content images was done as follows. For all possible $16 \times 16$ combinations of style and texture categories, exactly five cue conflict images were generated by randomly sampling style and content images from their respective categories. Sampling was performed without replacement for as long as possible, and then without replacement for the remaining images. The same stimuli acquired with this method were used for the cue conflict control experiments, where participants saw exactly these images but with different instructions biased towards shape and towards texture (results described later). For our analysis of texture vs. shape biases (Figure 4), we excluded trials for which no cue conflict was present (i.e., those trials where a bicycle content image was fused with a bicycle texture image, hence no texture-shape cue conflict present).

**Filled silhouette experiment** Style transfer is not the only possibility to generate a texture-shape cue conflict, and we here aimed at testing one other method to generate such stimuli: cropping texture images with a shape mask, such that the silhouette of an object and its texture constitute a cue conflict (visualised in Figure 7). Stimuli were generated by using the silhouette images from the

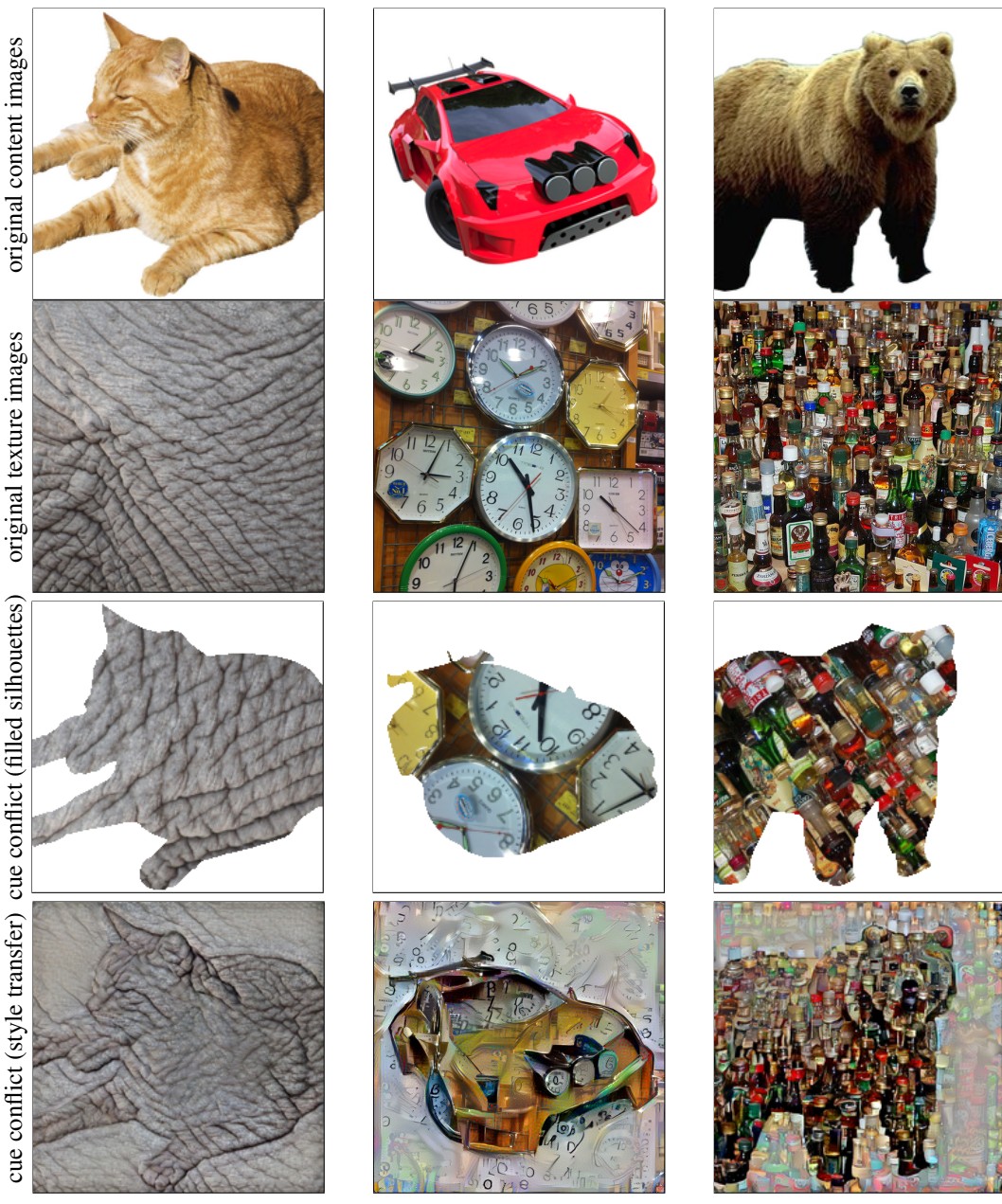

Figure 7: Visualisation of stimuli in data sets. Top two rows: content and texture images. Bottom rows: cue conflict stimuli generated from the texture and content images above (silhouettes filled with rotated textures; style transfer stimuli).

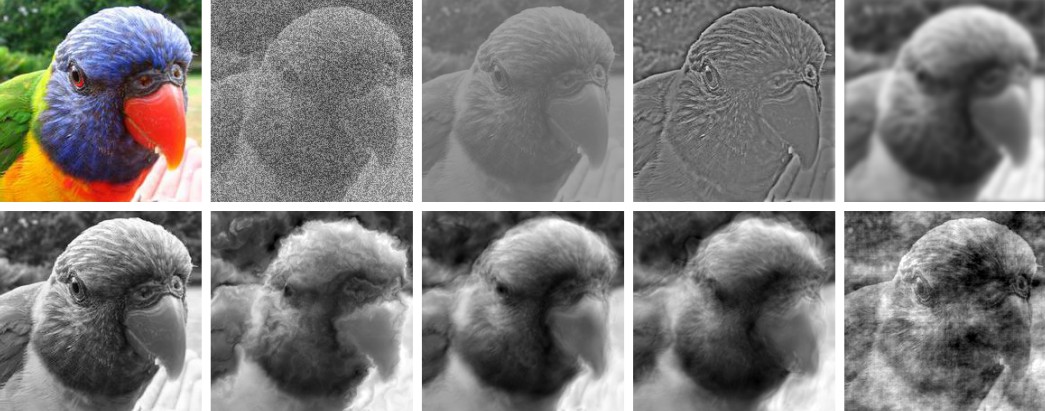

Figure 8: Visualisation of image distortions. One exemplary image (class `bird`, original image in colour at the top left) is manipulated as follows. From left to right: additive uniform noise, low contrast, high-pass filtering, low-pass filtering. In the row below, a greyscale version for comparison; the other manipulations from left to right are: Eidolon manipulations I, II and III as well as phase noise. Figure adapted from Geirhos et al. (2018) with the authors' permission.

silhouette experiment as a mask for texture images. If the silhouette image at a certain location has a black pixel, the texture was used at this location, and for white pixels the resulting target image pixel was white. In order to have a larger variety of textures than the 48 textures used in the texture experiment, the texture database was augmented by rotating all textures with ten different previously chosen angles uniformly distributed between 0 and 360 degrees, resulting in a texture database of 480 images. Results for this control experiment, not part of the main paper, are reported later. We ensured that no silhouette was seen more than once per observer.

**Robustness experiment (distorted images)**     For this experiment, human accuracies for reference were provided by Geirhos et al. (2018). Human 'error bars' indicate the full range of results for human observers. CNNs were then evaluated on different image manipulations applied to natural images as outlined in the paper. For maximal comparability, we also used the same images. For a description of the parametric distortion we kindly refer the reader to Geirhos et al. (2018). In Figure 8, we plot one example image across manipulations.

A.7    STYLIZED-IMAGENET (SIN)

We used AdaIN style transfer (Huang & Belongie, 2017) to generate Stylized-ImageNet. More specifically, the AdaIN implementation from `https://github.com/naoto0804/` `pytorch-AdaIN` (commit 31e769c159d4c8639019f7db7e035a7f938a6a46) was employed to stylize the entire ImageNet training and validation data sets. Style transfer was performed once per ImageNet image. As a style source, we used images from Kaggle's `Painter by Numbers` data set (`https://www.kaggle.com/c/painter-by-numbers/`, accessed on March 1, 2018). Style selection was performed randomly with replacement. Every ImageNet image was stylized once and only once. Paintings from the Kaggle data set were used if at least $224 \times 224$ pixels in size; the largest possible square crop was then downsampled to this size prior to using it as a style image. All accuracies are reported on the respective validation data sets. Code to generate Stylized-ImageNet from ImageNet (and the Kaggle paintings) is available on github in this repository: `https://github.com/rgeirhos/Stylized-ImageNet`

A.8    RESULTS: CUE CONFLICT CONTROL EXPERIMENTS (DIFFERENT INSTRUCTIONS)

We investigated the effect of different instructions to human observers. The results presented in the main paper for cue conflict stimuli correspond all to a neutral instruction, not biased w.r.t. texture or shape. In two separate experiments, participants were explicitly instructed to ignore the textures and click on the shape category of cue conflict stimuli, and vice versa. The results, presented in

| training | fine-tuning | top-1 IN accuracy (%) | top-5 IN accuracy (%) | Pascal VOC mAP50 (%) |
|---|---|---|---|---|
| IN (vanilla ResNet-152) | - | 78.31 | 94.05 | 76.9 |
| SIN | - | 65.26 | 86.31 | 75.0 |
| SIN+IN | - | 77.62 | 93.59 | 77.3 |
| SIN+IN | IN | **78.87** | **94.41** | **78.3** |

Table 4: Accuracy and object detection performance for ResNet-152. Accuracy comparison on the ImageNet (IN) validation data set as well as object detection performance (mAP50) on PASCAL VOC 2007. All models have an identical ResNet-152 architecture.

| training | ft | **mCE** | Noise | | | Blur | | | |
|---|---|---|---|---|---|---|---|---|---|
| | | | Gaussian | Shot | Impulse | Defocus | Glas | Motion | Zoom |
| IN (vanilla ResNet-50) | - | 76.7 | 79.8 | 81.6 | 82.6 | 74.7 | 88.6 | 78.0 | 79.9 |
| SIN | - | 77.3 | 71.2 | 73.3 | 72.1 | 88.8 | 85.0 | 79.7 | 90.9 |
| SIN+IN | - | **69.3** | **66.2** | **66.8** | **68.1** | **69.6** | **81.9** | **69.4** | 80.5 |
| SIN+IN | IN | 73.8 | 75.9 | 77.0 | 77.5 | 71.7 | 86.0 | 74.0 | **79.7** |

| training | ft | Weather | | | | Digital | | | |
|---|---|---|---|---|---|---|---|---|---|
| | | Snow | Frost | Fog | Brightness | Contrast | Elastic | Pixelate | JPEG |
| IN (vanilla ResNet-50) | - | 77.8 | 74.8 | 66.1 | 56.6 | 71.4 | 84.8 | 76.9 | 76.8 |
| SIN | - | 71.8 | 74.4 | 66.0 | 79.0 | **63.6** | 81.1 | 72.9 | 89.3 |
| SIN+IN | - | **68.0** | **70.6** | **64.7** | 57.8 | 66.4 | **78.2** | **61.9** | **69.7** |
| SIN+IN | IN | 74.5 | 72.3 | 66.2 | **55.7** | 67.6 | 80.8 | 75.0 | 73.2 |

Table 5: Corruption error (lower=better) on ImageNet-C (Hendrycks & Dietterich, 2019), consisting of different types of noise, blur, weather and digital corruptions. Abbreviations: mCE = mean Corruption Error (average of the 15 individual corruption error values); SIN = Stylized-ImageNet; IN = ImageNet; ft = fine-tuning. Results kindly provided by Dan Hendrycks.

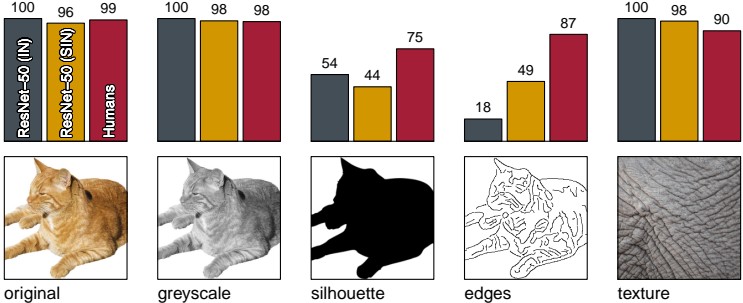

Figure 9: Accuracies and example stimuli for five different experiments without cue conflict, comparing training on ImageNet (IN) to training on Stylized-ImageNet (SIN).

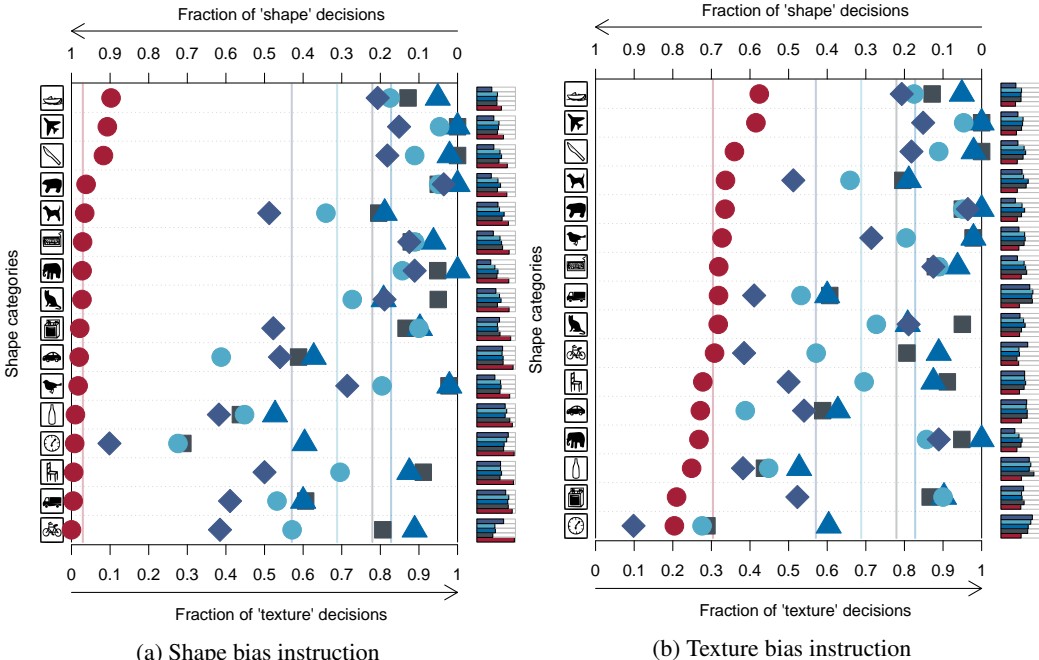

(a) Shape bias instruction

(b) Texture bias instruction

Figure 10: Classification results for human observers (red circles) and ImageNet-trained networks AlexNet (purple diamonds), VGG-16 (blue triangles), GoogLeNet (turquoise circles) and ResNet-50 (grey squares) on stimuli with a texture-shape cue conflict generated with style transfer, and *biased* rather than neutral instructions to human observers. Plotting conventions and CNN data as in Figure 4.

Figure 10, indicate that for a shape bias instruction, human data are almost exactly the same as for the neutral instruction reported earlier (indicating that human observers are indeed using shapes per default); and if they are instructed to ignore the shapes and click on the texture category, they *still* show a substantial shape bias (indicating that even if they seek to ignore shapes, they find it extremely difficult to do so).

## A.9 RESULTS: FILLED SILHOUETTE EXPERIMENT

This experiment was conducted as a control experiment to make sure that the strong differences between humans and CNNs when presented with cue conflict images are not merely an artefact of the particular setup that we employed. Stimuli are visualised in Figure 7; results in Figure 12. In a nutshell, we also find a shape bias in humans when stimuli are not generated via style transfer but instead through cropping texture images with a shape mask, such that the silhouette of an object and its texture constitute a cue conflict. CNNs have a less pronounced texture bias in these experiments;

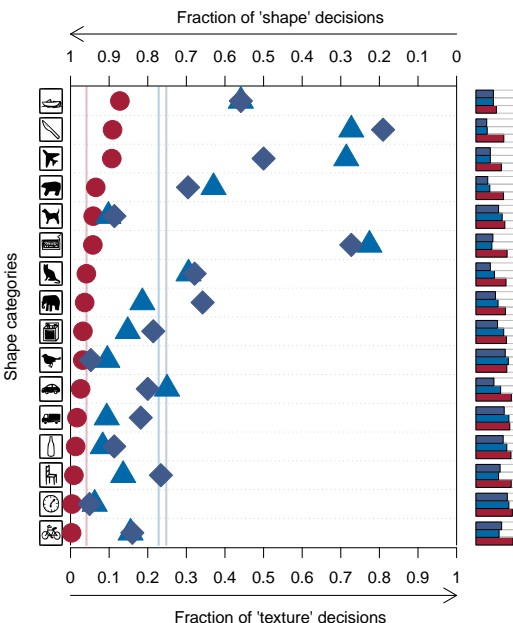

Figure 11: Texture vs shape biases on of AlexNet and VGG-16 after training on Stylized-ImageNet. Plotting conventions as in Figures 4 and 5. Plot shows biases for AlexNet (purple diamonds), VGG-16 (blue triangles) and human observers (red circles) for comparison. For GoogLeNet, no data is available since network training was performed in PyTorch and `torchvision.models` unfortunately does not provide a GoogLeNet (inception_v1) architecture.

ResNet-50 trained on SIN still responds with the shape category more than ResNet-50 trained on IN. Overall, these results are much more difficult to interpret since the texture-silhouette cue conflict stimuli, visualised in Figure 7, do not have a clear-cut texture-shape distinction like the cue conflict stimuli generated via style transfer. Still, they are largely in accord with the style transfer results presented in the main paper.

A.10  IMAGE RIGHTS & ATTRIBUTION

The images presented in Figure 7 were collected from different origins. We here indicate their URL, creator and license terms (if applicable). Some of the images presented in Figure 7 also appear in Figures 1, 2 and 9; the terms below apply accordingly. Top row, cat image: `https://pixabay.com/p-964343/`, released under the CC0 creative commons license as indicated on the website. The CC0 creative commons license is accessible from `https://creativecommons.org/publicdomain/zero/1.0/legalcode`. Car image: `https://pixabay.com/p-1930237/`, released under the CC0 creative commons license as indicated on the website. Bear image: ImageNet image `n02132136_871.JPEG`, manually modified to have a white background. Second row, elephant texture: cropped from `https://www.flickr.com/photos/flowcomm/5089601226`, released under the CC BY 2.0 license by user `flowcomm` as indicated on the website. The license is accessible from `https://creativecommons.org/licenses/by/2.0/legalcode`. Clock texture: cropped from `https://commons.wikimedia.org/wiki/File:` `HK_Sheung_Wan_%E4%B8%AD%E6%BA%90%E4%B8%AD%E5%BF%83_Midland_Plaza_` `shop_Japan_Home_City_clocks_displayed_for_sale_April-2011.jpg`, released under the Creative Commons Attribution-Share Alike 3.0 Unported, 2.5 Generic, 2.0 Generic and 1.0 Generic licenses by user `Ho Mei Danniel` as indicated on the website. The CC Attribution-Share Alike 3.0 license is accessible from `https://creativecommons.` `org/licenses/by-sa/3.0/legalcode`. Bottle texture: cropped from `https:` `//commons.wikimedia.org/wiki/File:Liquor_bottles.jpg`, released under the CC BY 2.0 license by user `scottfeldstein` as indicated on the website. The CC BY 2.0 license is accessible from `https://creativecommons.org/licenses/by/2.0/legalcode`.

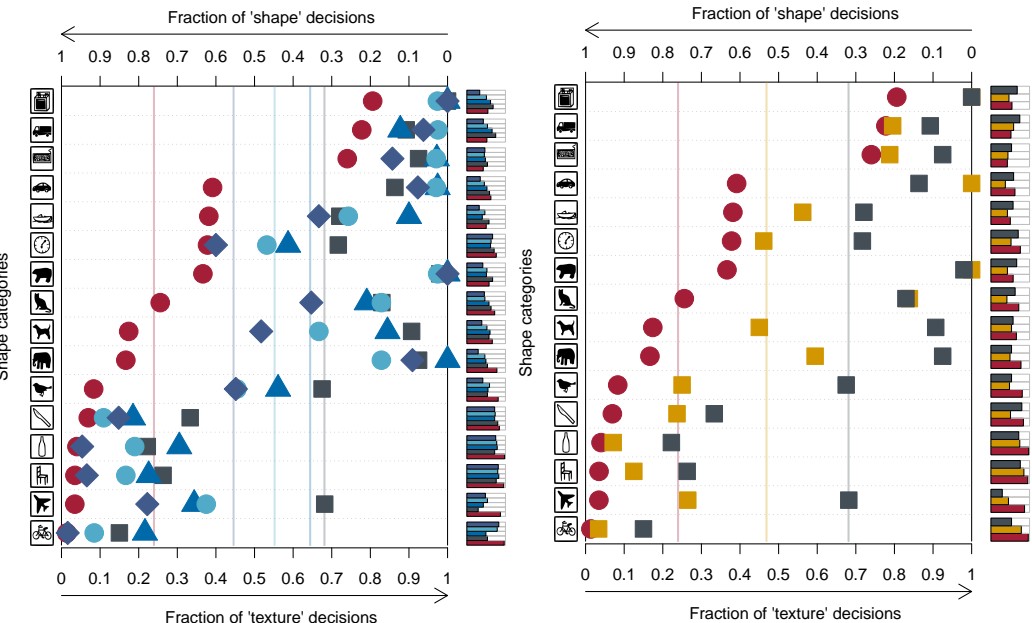

Figure 12: Classification results for human observers and CNNs on stimuli with a texture-silhouette cue conflict (filled silhouette experiment). Plotting conventions as in Figures 4 and 5.
**Left:** Human observers (red circles) and ImageNet-trained networks AlexNet (purple diamonds), VGG-16 (blue triangles), GoogLeNet (turquoise circles) and ResNet-50 (grey squares).
**Right:** Human observers (red circles, data identical to the left) and ResNet-50 trained on ImageNet (grey squares) vs. ResNet-50 trained on Stylized-ImageNet (orange squares).

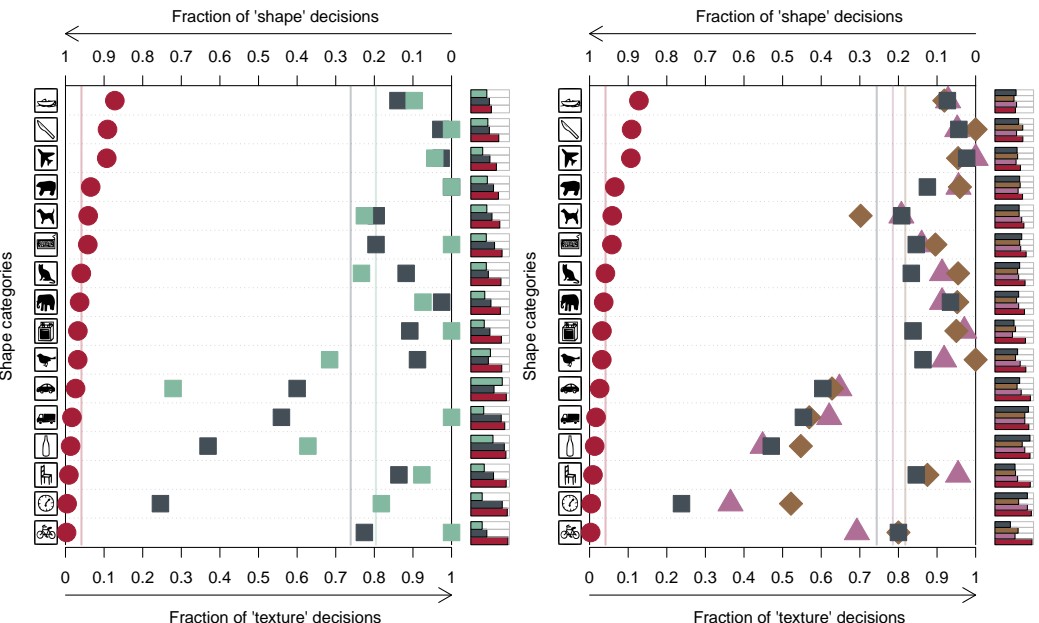

Figure 13: The texture bias on cue conflict stimuli is not specific to ImageNet-trained networks (left) and also occurs in very deep, wide and compressed networks (right).

**Left:** The texture bias is not specific to ImageNet-trained networks. Comparison of texture-shape biases on cue conflict stimuli generated with style transfer for ResNet-101 trained on ImageNet (grey squares) and ResNet-101 trained on the Open Images Dataset V2 (green squares) along with human data for comparison (red circles). Both networks have a qualitatively similar texture bias. We use a ResNet-101 architecture here since Open Images has released a pre-trained ResNet-101.

**Right:** The texture bias also appears in a very deep network (ResNet-152, grey squares), a very wide one (DenseNet-121, purple trianlges), and a very compact one (SqueezeNet1_1, brown diamonds). Human data for comparison (red circles). All networks are pre-trained on ImageNet.

