# OpenReview forum: "ImageNet-trained CNNs are biased towards texture; increasing shape bias improves accuracy and robustness"
_ICLR.cc/2019/Conference_

### Official Review · AnonReviewer1 · 2018-10-29
**Review of ImageNet-trained CNNs are biased towards texture**

**Rating:** 8
**Confidence:** 4

**Review:**

Review of ImageNet-trained CNNs are biased towards texture; increasing shape bias improves accuracy and robustness.

In this submission, the authors provide evidence through clever image manipulations and psychophysical experiments that CNNs image recognition is strongly influenced by texture identification as opposed the global object shape (as opposed to humans). The authors attempt to address this problem by using image stylization to augment the training data. The resulting networks appear much more aligned with human judgements and less biased towards image textures.

If the authors address my major concerns, I would increasing my rating 1-2 points.

Major Comments:

The results of this paper are quite compelling and address some underlying challenges in the literature on how CNN's function. I particularly appreciated Figure 5 demonstrating how the resulting stylized-augmented networks more closely align with human judgements. Additionally, it is surprising to me how poor BagNet performs on Stylized-ImageNet (SIN) implying that ResNet-50 trained on Stylized ImageNet may be better perceptually aligned with global object structure. Very cool.

1. Please make sure to tone down the claims in your manuscript. Although I share enthusiasm for your results, please recognize that stating that your results are 'conclusive' is premature and not appropriate. (Conclusive requires more papers and much work by the larger scientific community for a hypothesis to become readily accepted). Some sentences of concern include:

  --> "These experiments provide conclusive behavioural evidence in favour of the texture hypothesis"
  --> "we conclude the following: Textures, not object shapes, are the most important cues for CNN object recognition."

I would prefer to see language such as "We provide evidence that textures provide a more powerful statistical signal then global object shape for CNNs." or "We provide evidence that CNNs are overly sensitive to textures in comparison to humans perceptual judgements". This would be more measured and better reflect what has been accomplished in this study. Please do a thorough read of the rest of your manuscript and identify other text accordingly.

2. Domain shifts and data augmentation. I agree with your comment that domain shifts present the largest confound to Figure 2. The results of Geirhos et al, 2018 (Figure 4) indicate that individual image augmentations/distortions do not generalize well. Given these results, I would like to understand what image distortions were used in training each and all of your networks. Did you try a baseline with no image distortions (and/or just Stylized-ImageNet)?

Although the robustness in Figure 6 are great, how much of this can be attributed solely to Stylized-ImageNet versus the other types of image distortions/augmentations in each network. For instance, would contrast-insensitivity in Stylized-ImageNet diminish substantially if no contrast image distortion were used during training?

3. Semantics of 'object shape'. I suspect that others in the field of computer vision may take issue with your definition of 'object shape'. Please provide a crisp definition of what you test for as 'object shape' in each of your experiments (i.e. "the convex outline of object segmentation", etc.).

Minor Comments:

- Writing style in introduction. Rather then quoting phrases from individual papers, I would rather see you summarize their ideas in your own language and cite accordingly. This would demonstrate how you regard for their ideas and how these ideas fit together.

- Figure 2. Are people forced to select a choice or could they select 'I don't know'? Did you monitor response times to see if the manipulated images required longer times for individuals to pass decisions? I would expect that for some of the image manipulations that humans would have less confidence about their choices and that to be reflected in this study above and beyond an accuracy score.

- In your human studies, please provide some discussion about how you monitored performance to guard against human fatigue or lack of interest.

- Why did you use AdaIN instead of the original Gatys et al optimization method for image stylization? Was there some requirement/need for fast image stylization?

- Do you have any comment on the large variations in the results across class labels in Figure 4? Are there any easy explanations for this variation across class labels?

- Please use names of Shape-ResNet, etc. in Table 2.

- Are Pascal-VOC mAP results with fixed image features or did you fine-tune (back-propagate the errors to update the image features) during training? The latter would be particularly interesting as this would indicate that the resulting network features are better generic features as opposed to having used better data augmentation techniques.

- A.2. "not not used in the experiment" --> "not used in the experiment"

---

> ### Author Response · Authors · 2018-11-20
> **Author response to Reviewer 1**
>
> Dear Reviewer 1,
>
> Thank you for reviewing our paper, and for your helpful suggestions. We appreciate your assessment of our results as "very cool" and "surprising". We are happy to address your detailed suggestions and questions in a point-by-point response below.
>
> Writing: claims.
> We have addressed this concern, which was shared by Reviewer 2:
> https://openreview.net/forum?id=Bygh9j09KX&noteId=HygXM-Yb07
>
> Robustness and data augmentation.
> We made sure not to include any image distortions in the training data for any of our networks. Both models displayed in Figure 6 (ResNet-50 trained on ImageNet and ResNet-50 trained on Stylized-ImageNet) were trained under identical circumstances with respect to data augmentation (none apart from random resizing and flipping), hyperparameter settings, number of epochs, etc. Hence, any changes in the distortion robustness between these two models can be attributed solely to the changed training data (inducing different biases). We made this more clear in the Introduction as well as in Section 3.3 and the Discussion by writing that the SIN-trained network is more robust "despite never being trained on any of the distortions".
>
> Semantics of object shape.
> We have added the requested definitions in Section 2.2. We define "silhouette" as the bounding contour of an object in 2D (i.e., the outline of object segmentation). When mentioning "object shape", we use a definition that is broader than just the silhouette of an object: we refer to the set of contours that describe the 3D form of an object, i.e. including those contours that are not part of the silhouette.
>
> Quotes in Introduction.
> In general we agree that it is preferable to summarise ideas in one's own language. In this particular case, quotes are used with the aim to convince the reader that the "shape hypothesis" is not merely a straw man.
>
> Response times / forced choice.
> If human observers are allowed to select "I don't know", comparing results across participants with different confidence thresholds becomes very difficult. We therefore followed a standard psychophysical paradigm, namely an identification task with "forced choice" in the sense that they had to select a category even if unsure. However, observer confidence is typically correlated with reaction times (rapid responses for confident decisions), and we have thus added median reaction times across experiments as a column in Table 3 of the Appendix. This indeed shows that reaction times are longer for experiments with manipulated images.
>
> Guard against human fatigue / lack of interest.
> We have added the following explanation to Section A.4 of the Appendix:
> "Overall, we took the following steps to prevent low quality human data: 1., using a controlled lab environment instead of an online crowdsourcing platform; 2. the payment motivation scheme as explained above [i.e., better payment for better performance in experiments with a unique ground truth category]; 3. displaying observer performance on the screen at regular intervals during the practice session; and 4. splitting longer experiments into five blocks, where participants could take a break in between blocks."
>
> Reason for fast image stylization.
> We have added the following explanation to Section 2.3: "We used AdaIN fast style transfer rather than iterative stylization (e.g. Gatys et al., 2016) for two reasons: Firstly, to ensure that training on SIN and testing on cue conflict stimuli is done using different stylization techniques, such that the results do not rely on a single stylization method. Secondly, to enable processing entire ImageNet, which would take prohibitively long with an iterative approach."
>
> Explanation for large variations in results across labels.
> One possible explanation for the large variation in CNN results across labels may be that they use different strategies for different categories, i.e. that they sometimes rely solely on the texture (e.g. for category bear), and sometimes more on other cues. This is supported by the fact that there is a negative Spearman correlation between accuracy in our edge experiment, and texture bias in our cue conflict experiment (AlexNet: -0.582, GoogLeNet: -0.508, VGG-16: -0.238, ResNet-50: -0.014, human observers: -0.621): if a certain category seems hard to recognize from edges and contours, most networks are more likely to show a stronger class-conditional texture bias.
>
> Fine-tuning image features for Pascal VOC.
> We followed standard best practices for using image features in an object detection setting, which includes fine-tuning the image features. Importantly, this was done for all networks equally (which are trained on ImageNet and Stylized-ImageNet respectively), and the networks were trained under identical circumstances w.r.t. data augmentation. Thus, improved object detection performance can be attributed directly to better generic features induced by Stylized-ImageNet.
>
> Again, thank you very much for your review and suggestions!

---

> > ### Comment · AnonReviewer1 · 2018-11-26
> > **Response to rebuttal**
> >
> > I have read all of the other reviewer comments as well as the author responses to my original comments. The authors have addressed all of my primary concerns and clarified several issues that were not clear in the original manuscript. The additional results provided during the rebuttal (as requested by the other reviewers) provide even stronger evidence in favor of the central result. Based on this rebuttal and updates to the manuscript, I am upgrading my score from 6 to 8 as I think this will be an important piece of evidence in the design and analysis of deep network architectures for vision.

---

### Official Review · AnonReviewer2 · 2018-11-02
**Interesting idea and  nice demonstration**

**Rating:** 8
**Confidence:** 4

**Review:**

This paper talks about the behavior bias between human and advanced CNN classifier when classifying objects. A clear conclusion is that DNN classifiers lean on texture cues more than human, which is in contrast to empirical evidence. The experimental results are delighting and convincing to some extent. This paper is also inspiring and potentially useful to interpret how CNN works in object classification task.

Nevertheless, I have several small issues:
-	I like the writing of this paper, fluent description and clear topic. Besides, it provides sufficient information about experiment details, thus I think the experiments are fully reproducible. But I want to remind the authors to downplay their claims. Some sentences are not with academic rigor. i.e. “Textures, not object shapes, are the most important cues for CNN object recognition.”I don’t think it a good idea to claim textures as the “most important”cue.
-	Although adequate experiments are conducted on ResNet-50 on ImageNet, I miss experiments on a different object classification dataset i.e. PASCAL VOC, and a different network backbone such as very deep ResNet-152 or wider DenseNet. This lies in the concern that a different (deeper or wider) framework may behave quite differently and also the slightly shifted data distribution may induce controversial results. The adopted network ResNet-50, AlexNet, VGG-16 and GoogLeNet are not deep enough or either wide as DenseNet. Although transfer learning experiment is carried out upon PASCAL VOC, it’s not straightforward and not so truly telling. We’re curious about universal conclusions rather than that based on one dataset or network architecture of the same category. As a matter of fact, I’m nearly convinced by the provided results. But I think the demanding experiments will make the conclusions more solid.

Besides, I think the constructed dataset is beneficial to further research or fair comparison of future works, and I wonder the authors’ intention to publish such a dataset in the future.

I would raise my scores if the aforementioned problems are convincingly checked and solved.

---

> ### Author Response · Authors · 2018-11-20
> **Author response to Reviewer 2**
>
> Dear Reviewer 2,
>
> Thank you very much for your valuable feedback. We appreciate your assessment of our work as an "inspiring" paper. We provide a point-by-point response to your three suggestions below.
>
> Writing: claims.
> We have identified a number of sentences where our excitement about the results has biased our writing. We made the following changes:
> 1.) "the texture hypothesis: object textures, not object shapes as commonly assumed, are the most important cues for CNN object recognition." changed to "the texture hypothesis: in contrast to the common assumption, object textures are more important than global object shapes for CNN object recognition".
> 2.) "conclusive behavioural evidence" replaced with "behavioural evidence".
> 3.) In the Discussion, "we found strong evidence" replaced with "we provide evidence".
> 4.) In the Discussion, "we conclude the following: Textures, not object shapes, are the most important cues for CNN object recognition" replaced with "this highlights the special role that local cues such as textures seem to play in CNN object recognition".
> 5.) In the Summary, "we showed that machine recognition today primarily relies on texture rather than shape cues" replaced with "we provided evidence that machine recognition today overly relies on object textures rather than global object shapes as commonly assumed".
>
> Control experiments with different networks and data sets.
> You were expressing concern that deeper or wider networks and training on a different object classification data set may lead to different results. To address your concerns we have collected results for the requested control experiments. In Figure 13 of the Appendix, the left plot shows that the texture bias is equally prominent (if not stronger) in a network trained on the Open Images classification data set, thus the texture bias is not specific to training on ImageNet. The right plot of Figure 13 shows that ImageNet-trained networks ResNet-152 (a very deep network), DenseNet-121 (a very wide network) and for comparison also Squeezenet1_1 (a highly compressed network) all have a strong texture bias. Method details are reported in Section A.5 of the Appendix. Furthermore, we have started to train a ResNet-152 architecture on Stylized-ImageNet to use it as an additional "deep" backbone for our object detection experiments. Since network training and fine-tuning takes a lot of time, we will not be able to provide results by the end of the rebuttal period but we will make sure to include them afterwards. We have inquired with the ICLR organizers who have assured us it will be possible to make minor changes after the rebuttal period.
>
> Release of data sets.
> We are determined to release our cue conflict images (such as the cat with elephant skin) along with our raw data, image manipulation code, data analysis scripts, psychophysical experiment code and links to trained model weights in a github repository at the end of the anonymous review process. Furthermore, we will release code to create Stylized-ImageNet in a separate github repository along with a docker image; given two directory paths to ImageNet images (available from the ImageNet website [1]) and to the paintings used as a style source (available from Kaggle's painter-by-numbers website [2]) a shell script then creates Stylized-ImageNet.
>
> Again, thank you very much for reviewing our paper and for your valuable suggestions!
>
> [1] http://www.image-net.org/
> [2] https://www.kaggle.com/c/painter-by-numbers/data

---

### Official Review · AnonReviewer3 · 2018-11-02
**Interesting paper, purely empirical and no novelty**

**Rating:** 7
**Confidence:** 4

**Review:**

The paper is well written and easy to follow. It was a nice read for me.

The paper studies the CNNs like AlexNet, VGG, GoogleNet, ResNet50 and shows that these models are heavily biased towards the texture when trained on ImageNet. The paper shows human evaluations and compares model accuracies when various transformations like cue hypothesis, texture hypothesis (terms coined in the paper) are applied to study texture vs shape importance. The paper shows various results on different models clearly and results are easily interpretable. The paper then proposed a new ImageNet dataset which is called Stylized-ImageNet (SIN) where the texture is replaced with randomly selected painting style.

I believe that this is a good empirical study which is needed to understand why the ImageNet features are good (supervised training) and this can inform research in self-supervision, few shot learning domains.

The paper is an empirical paper and is presenting a quantitive study of role of texture which others have already presented like Gatys et al. 2017. The paper itself has no novel contributions. The paper notes "novel Stylized-ImageNet dataset" and shows that models can learn shape/texture features both but there is not much detail/explanation on why "Stylized" is the novel approach and also the methodology of constructing data by replacing with painting from AdaIN style transfer (Huang & Belongie, 2017) is not discussed/explored. More specifically, there is no ablation on other ways this dataset could have been constructed and why style transfer was picked as the choice, why was AdaIN chosen. While the choice is valid, I think these questions need to be answered if we have to consider it "novel". Additionally, I would like answers to the following questions:

1. In Figure 4, ResNet50 results are missing. I would be very interested in seeing those results. Can authors show those results?
2. Did authors study deeper networks like RN101/152 and do the observations about texture still hold?
3. Did authors consider inspecting if the models have same texture biases when trained on other datasets like COCO? If yes, can you share your results?
4. In Figure 5, can authors also show the results of training VGG, AlexNet, GoogleNet models on SIN dataset? I believe otherwise the results are incomplete since Fig. 4 shows the biases of these models on IN dataset but doesn't show if these biases are removed by training on SIN.
5. In Section 3.3, Transfer learning, authors show improvement on VOC 2007 Faster R-CNN . Do authors have explanation on why this gain happens? how's the texture learning in pretext task (like image classification training on SIN dataset) tied to the transfer learning no different dataset?
6. What are the results of transfer learning on other datasets like COCO, Faster R-CNN?

---

> ### Author Response · Authors · 2018-11-07
> **Clarification concerning novelty**
>
> Dear Reviewer 3,
>
> Thank you very much for reviewing our paper. Please allow us a quick and important clarification regarding your most important criticism ("The paper itself has no novel contributions") as we believe this may be due to a misunderstanding:
>
> First, we fully agree with you that one of the datasets we created, Stylized-ImageNet, is in itself not a major contribution - we use an existing fast style transfer method to strip ImageNet images of their original texture to replace it with the uninformative texture of a painting. Stylized-ImageNet is, for us, merely a means to an end, enabling us to make three (novel) core contributions:
>
> 1. Quantifying existing texture vs. shape biases.
> Many of the most influential explanations of CNN object recognition [1-3] describe it as a process of recognizing parts of objects / object shapes (the shape hypothesis). We contrast this with our carefully collected evidence for the texture hypothesis, offering an entirely different explanation. Furthermore, [e.g. 2,4-5] argue that CNNs closely mirror human object recognition and human shape perception. We here provide insights into a core difference of human and machine vision by comparing both under fair circumstances. To the best of our knowledge, our work is the first to systematically pitch shape against texture cues to investigate CNN biases and compare them to the human visual system.
>
> 2. Overcoming the texture bias in CNNs.
> Based on our texture hypothesis, we hypothesized that a CNN texture bias might be changed towards a shape bias if trained on a suitable dataset. We demonstrate the effectiveness of this approach, which shows that the texture bias in standard CNNs is not an inherent property of the architecture but rather induced by the training data. To the best of our knowledge, this is a novel finding and has never been attempted before. (The method of creating a suitable dataset, AdaIN style transfer, is not new.)
>
> 3. Showing emergent benefits of changed CNN biases.
> We demonstrate substantial advantages of a shape-based over a texture-based representation in CNNs, most importantly better features for transfer learning (object detection) and a previously unmatched robustness against a number of image distortions - despite never being trained on any of them. To the best of our knowledge, ShapeResNet is the first network to approach human-level distortion robustness on distortions that were not part of the training data.
>
> We believe that describing Stylized-ImageNet as "novel" - which we did in our original manuscript, e.g. in our abstract - was misleading since it is, as mentioned above and pointed out by you in your review, not a substantial contribution and, more importantly, merely a means to achieve our main and truly novel contributions. We will thus rephrase our description of Stylized-ImageNet throughout the paper to avoid any misunderstandings by future readers who might get the idea that the dataset itself is being described as our core novel contribution. Apologies for any confusion this issue may have caused in the original submission.
>
> We would appreciate it if you could let us know whether this clarifies the issue, and changes your assessment of the novelty of our work.
>
> [1] Goodfellow, I., Bengio, Y., Courville, A., & Bengio, Y. (2016). Deep learning (Vol. 1). Cambridge: MIT press.
> [2] Kriegeskorte, N. (2015). Deep neural networks: a new framework for modeling biological vision and brain information processing. Annual review of vision science, 1, 417-446.
> [3] LeCun, Y., Bengio, Y., & Hinton, G. (2015). Deep learning. Nature, 521(7553), 436.
> [4] Kubilius, J., Bracci, S., & de Beeck, H. P. O. (2016). Deep neural networks as a computational model for human shape sensitivity. PLoS computational biology, 12(4), e1004896.
> [5] Cadieu, C. F., Hong, H., Yamins, D. L., Pinto, N., Ardila, D., Solomon, E. A., ... & DiCarlo, J. J. (2014). Deep neural networks rival the representation of primate IT cortex for core visual object recognition. PLoS computational biology, 10(12), e1003963.

---

> ### Author Response · Authors · 2018-11-20
> **Author response to Reviewer 3**
>
> Dear Reviewer 3,
>
> Thank you for your review and feedback. We appreciate your assessment of our work as a "good empirical study" and "well-written" paper. We provide a point-by-point response to individual concerns below.
>
> 1. ResNet-50 results not included in Figure 4.
> Thank you for pointing this out. Originally, results for ResNet-50 were included in Figure 5 but not in Figure 4. We have now added them to Figure 4 as well (and likewise, to Figure 2).
>
> 2. Does the texture bias exist in deeper networks like ResNet-152?
> We have addressed this interesting question by investigating the texture bias in three additional networks: a very deep network (ResNet-152), a very wide network (DenseNet-121) and a highly compressed network (SqueezeNet1_1). The results are reported in the Appendix, Figure 13 (right). All three networks show a strong texture bias.
>
> 3. Texture bias when trained on a different data set.
> We appreciate this suggestion and validated our results on a different training data set. We have investigated the texture bias of a ResNet-101 architecture trained on a different data set, namely Open Images. In Figure 13 of the Appendix, the left plot shows that the texture bias is equally prominent (if not stronger) in a network trained on the Open Images classification data set, thus the texture bias is not specific to training on ImageNet.
>
> 4. Training other models on Stylized-ImageNet.
> We have additionally trained VGG-16 and AlexNet on Stylized-ImageNet (SIN) as suggested. The results are available in the Appendix, Figure 11 and linked from the caption of Figure 5 (in which we only show results for one network, ResNet-50, in order to avoid a cluttered plot). Training on SIN does remove the texture bias in these other networks just like it does remove it for ResNet-50.
>
> 5. Explanation for improved object detection performance.
> We have added the following sentence to the "transfer learning" section: "This [the improved detection performance] is in line with the intuition that for object detection, a shape-based representation is more beneficial than a texture-based representation, since the ground truth rectangles encompassing an object are by design aligned with global object shape."
>
> 6. Transfer learning on different object detection data set.
> We believe that validating our transfer learning results on a different data set (MS COCO) is an important suggestion. We are working towards including these results as an additional column in Table 2. We will not be able to provide results by the end of the rebuttal period but we will make sure to include them once completed. We have inquired with the ICLR organizers who have assured us it will be possible to make minor changes after the rebuttal period.
>
> Novelty.
> We have written a fast response to this point ("Clarification concerning novelty": https://openreview.net/forum?id=Bygh9j09KX&noteId=S1g75tNxTQ ) in the hope that this clarifies the issue. We have identified the following sentences that were misleading in our original submission, and we have changed them as described below to avoid any misunderstandings by future readers who might get the idea that the data set itself is being described as our core novel contribution:
> a) In the abstract, we replaced "our novel Stylized-ImageNet dataset" with "Stylized-ImageNet, a stylized version of ImageNet".
> b) In the Introduction, we replaced "Utilising style transfer (Gatys 2015), we devised a novel way to create images with a texture-shape cue conflict" with "Utilising style transfer (Gatys 2015), we created images with a texture-shape cue conflict".
> c) We adapted the last paragraph of our Introduction such that our contributions, as we see them, are explicitly mentioned ("Beyond quantifying existing biases, we subsequently present results for our two other main contributions: changing biases, and discovering emergent benefits of changed biases.").
> d) and e) In Section 3.2 and in the Summary, we replaced "a novel Stylized-ImageNet (SIN) data set" with "our Stylized-ImageNet (SIN) data set".
>
> Again, thank you very much for reviewing our paper and for your valuable suggestions!

---

### Public Comment · ~Grigorios_Chrysos1 · 2018-10-15
**Nice work + questions**

Great idea to scrutinize the shape hypothesis.

However, as this work is breaking some strong assumptions of the community, the following points were not clear:

1)  Were the networks trained to recognize only the 16 classes or the typical 1,000 of imagenet? If the latter, how was a random prediction restricted to the 16 classes?
2) How were these categories selected? They have both distinct appearance and texture in most cases.

Once again, very promising work from the authors, looking forward for the code and implementations.

---

> ### Author Response · Authors · 2018-10-29
> **Author response to public comment "Nice work + questions"**
>
> Thank you for your interest in our work, and for your comment!
> We are happy to clarify the questions, and will make these points more clear in the next version of our paper.
>
> 1) All networks were trained on full ImageNet (or full Stylized-ImageNet); they thus recognize 1,000 classes. Concerning the mapping from 1,000 classes to 16 classes, we followed the procedure introduced in [1] (the paper where 16-class-ImageNet was proposed). In order to achieve a fair comparison to the forced-choice paradigm for human observers (who were given a choice of 16 categories on the lab response screen), only those ImageNet categories corresponding to one of the 16 entry-level categories were considered for the network response. The mapping between ImageNet and 16-class-ImageNet categories was achieved via the WordNet hierarchy [2] - e.g. ImageNet category "tabby cat" would be mapped to "cat".
>
> 2) We used the 16-class-ImageNet categories introduced in [1]. These are the 16 entry-level categories from MS COCO that have the highest number of ImageNet classes mapped to them via the WordNet hierarchy.
>
> [1] Geirhos, Temme, Rauber, Schütt, Bethge, Wichmann: Generalisation in humans and deep neural networks, NIPS 2018, https://arxiv.org/pdf/1808.08750.pdf
> [2] George A Miller. Wordnet: a lexical database for English. Communications of the ACM, 38(11):39–41, 1995.

---

> > ### Public Comment · ~Grigorios_Chrysos1 · 2018-11-01
> > **Thanks**
> >
> > Thanks for the reply.

---

### Public Comment · (anonymous) · 2018-11-19
**Stylized-ImageNet Details**

When constructing Stylized-ImageNet, does content-style trade-off coefficient for AdaIN vary? If not, what is the coefficient? Does training with several stylizations per image work better than just one stylization per image?
Also could the authors show that networks trained with SIN generalize to the set of corruptions from https://openreview.net/forum?id=HJz6tiCqYm so that we know that the distortions are not cherry-picked?

---

> ### Author Response · Authors · 2018-11-19
> **Author response to public comment "Stylized-ImageNet Details"**
>
> Thanks for your interest in our work!
>
> As mentioned in the paper, we used the PyTorch implementation from [1]. The degree of stylization (parameter "alpha" in the implementation) was kept at the default value of 1.0; it might be interesting to explore whether a lower coefficient still nudges a model towards a shape bias.
>
> We used training with one stylization per image since this allows to pre-process ImageNet once rather than on-the-fly, which is desirable for faster training. In principle, our approach enables up to 79,434 different stylizations of a single image (this is the size of our style dataset), and we would expect that using more stylizations leads to even better results in terms of both SIN and IN accuracy.
>
> Once the anonymous review period ends, we will release all of our trained model weights to facilitate comparisons to other models and other data sets like the one mentioned in your comment.
>
> [1] https://github.com/naoto0804/pytorch-AdaIN

---

> > ### Public Comment · ~Dan_Hendrycks1 · 2018-12-22
> > **Results Are Not Cherry-Picked**
> >
> > Using the 75 corruptions from ImageNet-C, a ResNet-50 with Stylized ImageNet data augmentation has its corruption robustness error rate decrease from 76.70% to 69.32%, which is a very substantial improvement. Moreover, the network also did better on most types of blurs, so the current draft's negative finding that Stylized ImageNet can hurt performance for blur corruptions is a product of not testing on a broader range of blurs. When more types of blurs are tested, Stylized ImageNet data augmentation proves overall beneficial in that case too.

---

> > > ### Author Response · Authors · 2018-12-29
> > > **Incorporating results on ImageNet-C**
> > >
> > > Thanks for reaching out via Email and providing the results of a comprehensive analysis on ImageNet-C, very much appreciated. Following our Email exchange, I have uploaded an updated version incorporating these additional results on ImageNet-C image corruptions for ResNet-50 with various degrees of Stylized-ImageNet training (Table 4 of the Appendix + summary in main text, Section "robustness against distortions").

---

### Author Response · Authors · 2018-11-26
**Author's summary of rebuttal discussion**

We would like to thank all reviewers for their valuable feedback and we very much appreciate their assessment of our work as "surprising" & "very cool" (R1), "inspiring" (R2) and "well-written" (R3).

This is a summary of main concerns and how we addressed them.

- additional control experiments (R2, R3): We have conducted the requested control experiments (texture bias in wider & deeper networks and different training dataset). Results consistently support original findings.

- rephrasing some claims (R1, R2): we provided a listing of changed statements in the detailed rebuttals.

- novelty (R3): we believe this is due to a misunderstanding; we have written a clarification and stated our contributions more clearly.

- further clarifications (R1), release of dataset (R2), plotting improvements (R3), etc.: we have addressed all of them and updated the paper accordingly.

R1, R2 and R3 have already taken the time to assess our changes and indicated being happy with the updated version.

---

> ### Author Response · Authors · 2019-02-04
> **Update: new object detection results on MS COCO and with "deep" backbone**
>
> The reviewers mentioned that it would be useful to complement our results with additional control experiments, which we have now included:
>
> 1. Object detection on a different dataset (MS COCO) with a model trained on Stylized-ImageNet (SIN)
> -> Results now included in Table 2. Training on SIN substantially improves detection performance from 52.3 to 55.2 mAP50.
>
> 2. Object detection using a deeper architecture
> -> Results now included in Table 4 of the Appendix. Training a ResNet-152 on SIN substantially improves detection performance on PASCAL VOC 2007 from 76.9 to 78.3 mAP50. ImageNet validation accuracies are also improved in comparison to a vanilla ImageNet-trained ResNet-152.
>
> Taken together, these results highlight that incorporating SIN in the training data consistently leads to better features for object detection.

---

### Public Comment · (anonymous) · 2022-10-18
**Dataset size when comparing IN and SIN**

Very interesting paper.
One point is that when training on IN+SIN the size of data set is doubled in comparison by when training on IN. I am suspicious that it might be part of reason behind improved results for IN+SIN.

---

### Meta-Review · Area_Chair1 · 2018-12-16
**Area chair recommendation**

**Confidence:** 5
**Recommendation:** Accept (Oral)

**Metareview:**

This paper proposes a hypothesis about the kinds of visual information for which popular neural networks are most selective.  It then proposes a series of empirical experiments on synthetically modified training sets to test this and related hypotheses.  The main conclusions of the paper are contained in the title, and the presentation was consistently rated as very clear.  As such, it is both interesting to a relatively wide audience and accessible.

Although the paper is comparatively limited in theoretical or algorithmic contribution, the empirical results and experimental design are of sufficient quality to inform design choices of future neural networks, and to better understand the reasons for their current behavior.

The reviewers were unanimous in their appreciation of the contributions, and all recommended that the paper be accepted.